# Reaction Sintering of MgAlON at 1500 °C from Al$_2$O$_3$, MgO and AlN and Its Wettability by AlSi7Mg

**Alina Schramm** [1,*], **Martin Thümmler** [2], **Olga Fabrichnaya** [2], **Simon Brehm** [3], **Jakob Kraus** [3], **Jens Kortus** [3], **David Rafaja** [2], **Christiane Scharf** [1,*] and **Christos G. Aneziris** [4]

1   Institute for Nonferrous Metallurgy and Purest Materials, Technische Universität Bergakademie Freiberg, Leipziger Straße 34, 09599 Freiberg, Germany

2   Institute of Materials Science, Technische Universität Bergakademie Freiberg, Gustav-Zeuner-Straße 5, 09599 Freiberg, Germany; martin.thuemmler@iww.tu-freiberg.de (M.T.); fabrich@ww.tu-freiberg.de (O.F.); david.rafaja@ww.tu-freiberg.de (D.R.)

3   Institute of Theoretical Physics, Technische Universität Bergakademie Freiberg, Leipziger Straße 23, 09599 Freiberg, Germany; simon.brehm@physik.tu-freiberg.de (S.B.); jakob.kraus@physik.tu-freiberg.de (J.K.); jens.kortus@physik.tu-freiberg.de (J.K.)

4   Institute of Ceramics, Refractories and Composite Materials, Technische Universität Bergakademie Freiberg, Agricolastr. 17, 09599 Freiberg, Germany; aneziris@ikfvw.tu-freiberg.de

*   Correspondence: alina.schramm1@ikgb.tu-freiberg.de (A.S.); christiane.scharf@inemet.tu-freiberg.de (C.S.)

**Abstract:** The aim of this study is the investigation of a technological synthesis of MgAlON, which is a prospective coating material on ceramic foam filters for the filtration of magnesium, aluminum, and other metal melts. Thermodynamic calculations are performed, and the synthesis is carried out at 1500 °C in nitrogen atmosphere using samples consisting of different fractions of Al$_2$O$_3$, MgO, and AlN as starting materials. The effect of the quantity of these components on the conversion degree of the educts is evaluated. Furthermore, the effect of the holding time at the synthesis temperature, as well as the composition points or regions showing the highest conversion degree, are determined. XRD analysis is performed to evaluate the phase fractions and lattice parameters of the spinel after the respective reaction, and the nitrogen content of selected samples is evaluated. Sessile drop tests using AlSi7Mg are performed at 950 °C on selected sintered samples, determining their wettability, and therefore, applicability of the material in light metal melt filtration.

**Keywords:** MgAlON; reaction sintering; nitride ceramics; wettability; refractory

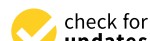



## 1. Introduction

Magnesium-containing AlON compounds were described for the first time in 1976 by Jack et al. [1], who synthesized various nitrogen-containing ceramics. Magnesium aluminum oxynitride (MgAlON) is understood as a solid solution of γ-AlON and magnesium aluminate spinel (Mg$_x$Al$_{(8-2x)/3}$O$_4$) [2], having a defect spinel structure (space group: Fd$\bar{3}$m) [3]. The γ-AlON phase is stable at temperatures above 1640 °C [4], melts congruently at 2165 °C, and covers a homogeneity range from 28 to 40 mol% AlN [5]. MgAl$_2$O$_4$ is stable down to low temperatures; its addition stabilizes AlON at temperatures below 1640 °C [6]. At temperatures above 1640 °C γ-AlON and magnesium aluminate spinel form a continuous solid solution [7,8]. MgAlON exhibits high strength, high thermal and corrosion resistance, and good thermal shock resistance [9,10]. Bending strengths up to 25 MPa at 1000 °C were reached in MgAlON spinel composites [11]. Dai et al. [10] reported that MgAlON powder starts to oxidate and therefore degrade gradually above 750 °C. High-speed oxidation of MgAlON bulk material takes place at temperatures above 1600 °C. The reaction products are alumina, magnesium aluminate, and nitrogen [10]. Below 1200 °C, MgAlON shows the best oxidation resistance among other ceramic refractory materials, such as AlON, SiAlON-ZrO$_2$, and BN-ZCM [12].

As it is not easily wetted and eroded by oxidic slags, MgAlON has been investigated as a promising refractory material for steelmaking applications [13].

Furthermore, MgAlON is regarded as a promising material for magnesium melt filtration, since the temperature regime of magnesium alloy casting is commonly held below 700 °C. Preliminary immersion tests of ceramic foam filters in the molten magnesium alloy AZ91 were previously conducted at 680 °C [14,15]. Still, it should be noted that processing conditions required for MgAlON synthesis, such as high temperatures and requirements towards the purity of the synthesis atmosphere, hinder its wide industrial application [16].

Several different techniques have been reported for the synthesis of MgAlON, such as pressureless reaction sintering using $Al_2O_3$, MgO, and AlN as starting materials [9,13,17–19]; carbothermal reduction-nitridation, which uses $Al_2O_3$, MgO, and a carbon source under pure nitrogen [20]; and aluminothermal reduction in pure nitrogen atmosphere, for which $Al_2O_3$, MgO, and metallic Al powder are used as starting materials [2]. Other applied techniques include spark plasma sintering [21,22] and hot pressing [7].

According to Gao et al. [2], MgAlON begins to be formed at 1165 °C during pressureless reaction sintering, although residues of AlN and $Al_2O_3$ phases remain if the sintering temperature is below 1600 °C. Cheng et al. [13] reported MgAlON as the major phase at 1500 °C, with a small portion of $Al_2O_3$ and AlN remaining, whereas the starting material MgO is entirely used up during the five-hour reaction sintering process in nitrogen atmosphere. To reach this result, a mixture of 78.74 wt% $Al_2O_3$, 8.53 wt% aluminum powder, and 12.73 wt% MgO was used. It was furthermore concluded that the temperature required for the formation of single-phase MgAlON rises with rising AlN-content of the mixture [13]. Experiments within the $Al_2O_3$-AlN-MgO system conducted by Granon et al. [23] yielded the formation of single-phase MgAlON after a thermal treatment at 1400 °C for 3 h for mixtures containing 57 mol% $Al_2O_3$, 39 or 21 mol% MgO, and 4 or 22 mol% AlN, (see Table 1 and Figure 1).

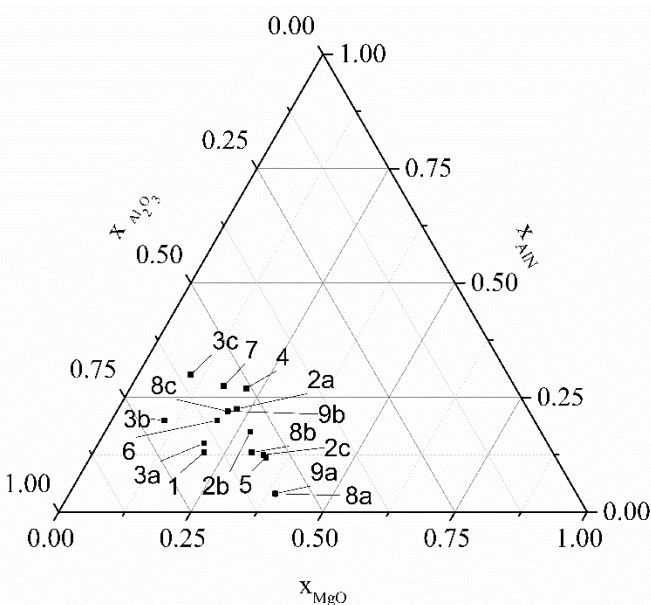

**Figure 1.** Starting compositions (mole fractions) for the synthesis of MgAlON from selected literature sources, specified in Table 1, shown in a simplified diagram of the ternary MgO-AlN-$Al_2O_3$ system.

Ma et al. [24] also reported MgAlON as the major phase after 2 h of pressureless sintering of a carbothermal mixture of 3.6 wt% carbon black, 88.6 wt% $Al_2O_3$, and 7.8 wt% MgO in flowing nitrogen at 1500 °C, whereas single-phase MgAlON was observed at 1600 °C. For an aluminothermal reduction using 51 mol% $Al_2O_3$, 22 mol% MgO, and 27 mol% aluminum powder as starting materials, Wang et al. [25] reported nearly single-phase MgAlON after holding the mixture at 1500 °C for 3 h in flowing nitrogen.

**Table 1.** Starting compositions (mole fractions) of AlN, MgO, and $Al_2O_3$ used in literature for reactive sintering of MgAlON.

| Number (cf. Figure 1) | Source | $x_{AlN}$ | $x_{MgO}$ | $x_{Al_2O_3}$ | Synthesis Temperature, Duration |
|---|---|---|---|---|---|
| 1 | [18] | 0.13 | 0.21 | 0.66 | 1550 °C, 6 h; 1675 °C, 6 h |
| 2a | [13] | 0.225 | 0.225 | 0.55 | 1500–1650 °C, 5 h |
| 2b | [13] | 0.175 | 0.275 | 0.55 | 1500–1650 °C, 5 h |
| 2c | [13] | 0.125 | 0.325 | 0.55 | 1500–1650 °C, 5 h |
| 3a | [26] | 0.15 | 0.2 | 0.65 | 1550–1800 °C, 3 h |
| 3b | [26] | 0.2 | 0.1 | 0.7 | 1550–1800 °C, 3 h |
| 3c | [26] | 0.3 | 0.1 | 0.6 | 1550–1800 °C, 3 h |
| 4 | [25] | 0.27 | 0.22 | 0.51 | 1300–1600 °C, 3 h |
| 5 | [16] | 0.119 | 0.333 | 0.548 | 1550–1650 °C, 5 h |
| 6 | [27] | 0.2 | 0.2 | 0.6 | Hot pressing, SPS, 1600 °C, 1 h |
| 7 | [19] | 0.275 | 0.175 | 0.55 | 1450–1700 °C, 6 h |
| 8a | [3] | 0.04 | 0.39 | 0.57 | 1450 °C, 9 h |
| 8b | [3] | 0.13 | 0.3 | 0.57 | 1450 °C, 9 h |
| 8c | [3] | 0.22 | 0.21 | 0.57 | 1450 °C, 9 h |
| 9a | [23] | 0.04 | 0.39 | 0.57 | 1400 °C, 3h |
| 9b | [23] | 0.22 | 0.21 | 0.57 | 1400 °C, 3h |

Figure 1 shows an overview of compositions gathered from the literature in which researchers had worked with similar temperature regimes and raw materials to those planned for this research work and succeeded in synthesizing MgAlON.

The aim of this study was to synthesize MgAlON from mixtures of $Al_2O_3$, AlN, and MgO by means of reactive sintering, following an experimental simplex layout within the promising area of the pseudo-ternary phase diagram.

The effects of the starting composition and preparation route of the respective mixture as well as the holding time at synthesis temperature on the yield of the synthesis are to be investigated. These points were perceived as a research gap in previous studies.

In future research, the synthesized MgAlON powder is to be used for the coatings of carbon-free or carbon-bonded alumina filters, which will in turn be applied in filtration experiments with metal melts, especially magnesium [14,15] and aluminum alloys.

## 2. Theoretical and Thermodynamic Considerations

For the description of the chemical composition and structure of MgAlON, a wide range of modified chemical formulas is used in the literature [7,10,23,26–28]. Furthermore, the depiction of the composition within the pseudo-ternary system $MgO$-$AlN$-$Al_2O_3$ is well established. At temperatures over 1400 °C, the MgAlON phase extends over a relatively large homogeneity range [4,13,23,29]. This includes the ideal stoichiometric spinel $MgAl_2O_4$ $(x_{MgO} = x_{Al_2O_3} = 0.5, x_{AlN} = 0)$, which is also a part of the pseudo-binary subsystem $MgO$-$Al_2O_3$. In its equilibrium state at ambient conditions, the unperturbed crystal structure of $MgAl_2O_4$ is described by the space group $Fd\bar{3}m$, where the anions $O^{2-}$ occupy the Wyckoff sites $32e$, forming an approximately face-centered cubic (fcc) lattice. Within this anion sublattice, the bivalent cations $Mg^{2+}$ and the trivalent cations $Al^{3+}$ occupy the Wyckoff sites $8a$ (tetrahedral positions) and $16d$ (octahedral positions), respectively. However, with increasing temperature, the inversion occurs, which means that $Al^{3+}$ cations occupy tetrahedral sites (inversion degree) whereas $Mg^{2+}$ cations occupy octahedral sites without a change in the stoichiometry of $MgAl_2O_4$. At high temperatures, the homogeneity range of MgAlON extends towards $Al_2O_3$ as well as towards MgO (see phase diagram shown in [30,31]). Therefore, the description of the spinel structure in the $MgO$-$Al_2O_3$ system $(x_{MgO} \neq x_{Al_2O_3}, x_{AlN} = 0)$ requires either the incorporation of vacancies or interstitials on cation sites different from $8a$ and $16d$ [28]. It should be noted that $Al_2O_3$ solubility in spinel increases with temperature and can reach ≈88 mol% $Al_2O_3$. The investigation of spinel containing excess $Al_2O_3$ indicated the formation of vacancies on cationic sites [32], which

allows electroneutrality to be maintained. In this specific case, the amount of cation vacancies reached a maximum of approximately 9.2% ($Mg_{0.175}Al_{2.55}V_{0.275}O_4$), which prefers the occupation of octahedral sites [33]. Moderate MgO solubility in spinel was suggested to model the occupation of interstitial sites by $Mg^{2+}$ cations [30].

In the $\gamma$-AlON spinel, the substitution of $O^{2-}$ by $N^{3-}$ anions occurs on 32*e* sites. Hetero-valent substitution is possible only when the ratio of $O^{-2}$ to $N^{-3}$ is equal to 3:1 to keep electroneutrality. Fractions of $O^{-2}$ and $N^{-3}$ in anionic sites are equal to 0.75 and 0.25, respectively, which correspond to a spinel composition of 50 mol% AlN and 50 mol% $Al_2O_3$. However, according to a phase diagram of the $Al_2O_3$-AlN system, AlON with a spinel structure forms at compositions substantially enriched by $Al_2O_3$ (>60 mol%) in comparison with this ideal composition of 1:1. Therefore, the formation of vacancies in cationic sites is necessary to keep electroneutrality, predominantly in octahedral sites [34,35]. Around 2500 °C the $Al_2O_3$ solubility is predicted to reach a maximum of $\approx$94 mol% $Al_2O_3$, resulting in approximately 10.2% cation vacancies ($Al_{2.69}V_{0.31}O_{3.92}N_{0.08}$) [36].

To perform calculations in the MgO-$Al_2O_3$-AlN system, the thermodynamic database SGSUB (SGTE) [37] was used. It should be noted that spinel is modeled as a stoichiometric compound ($MgAl_2O_4$). The AlON spinel was not considered, as it is not stable below 1594 °C [36] and would therefore not be formed at the present experimental conditions. The database of the $Al_2O_3$-AlN system was created by combining the thermodynamic description of Dumitrescu and Sundman [38] for solid and liquid phases, with the description of the gas phase accepted from the SGSUB (SGTE) database.

$MgAl_2O_4$ spinel is stable from low temperatures up to its melting point at 2100 °C [31]. In the Mg-Al-O-N system, $MgAl_2O_4$ spinel can be in equilibrium with the gas atmosphere, AlN, and with the respective oxide, $Al_2O_3$ or MgO. In the case of equilibrium of spinel, AlN, $Al_2O_3$, and gas, a major constituent of gas is $N_2$, followed by Mg. However, the next constituent with the highest content in the gas phase after Mg is either Al, in a temperature range from 1000 to 1531 °C, or $Al_2O$, in the range of 1531 to 2000 °C (see Figure 2a). Calculated activities of major gas species are presented in Figure 2a. The partial pressure of oxygen increases with temperature from a value of $P(O_2)$ equal to $1.26 \times 10^{-25}$ bar at 1000 °C to a value of $2.51 \times 10^{-13}$ bar at 2000 °C (see Figure 2b).

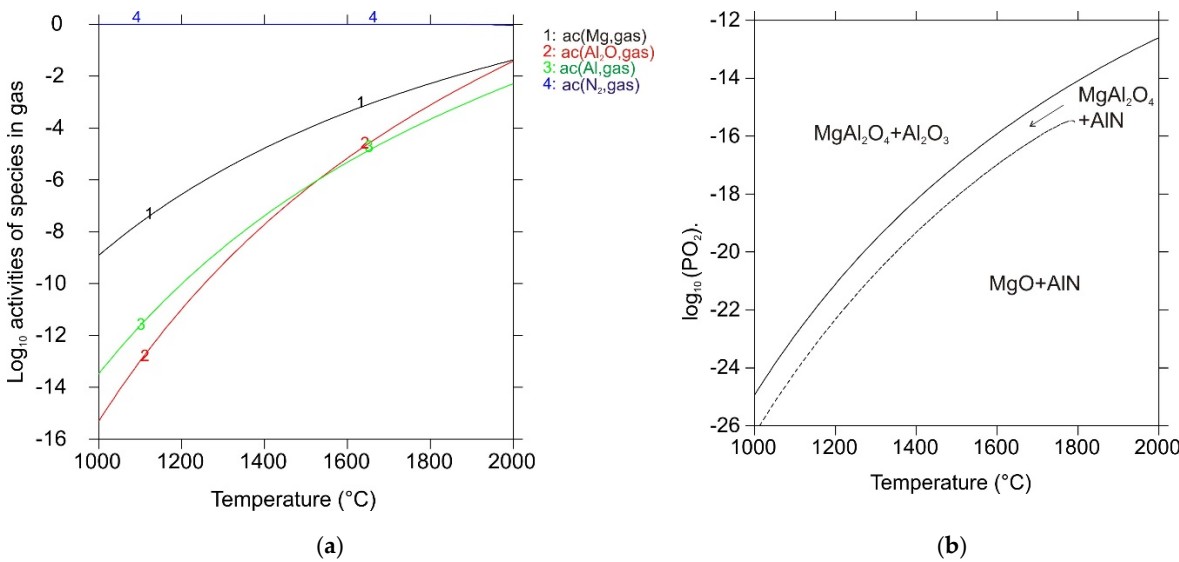

(a)                                              (b)

**Figure 2.** Equilibrium of $MgAl_2O_4$ with $Al_2O_3$ and AlN in the presence of gas as a function of temperature: (**a**) activities of gas species and (**b**) partial pressure of oxygen (in bar); the dashed line shows the stability limit of $MgAl_2O_4$.

At an oxygen partial pressure above the equilibrium line, AlN is not stable and $MgAl_2O_4$ coexists with $Al_2O_3$, whereas $MgAl_2O_4$ coexists with AlN at an oxygen partial pressure below the equilibrium line. The calculated temperature dependence of the oxygen

partial pressure for the equilibrium of MgAl$_2$O$_4$ with Al$_2$O$_3$, AlN, and gas is presented by the solid line in Figure 2b.

Calculations of the equilibrium of MgAl$_2$O$_4$ spinel with gas, MgO, and AlN (see Figure 3) indicated a substantially higher content of Mg in the gas phase than in the previously calculated case. The concentration of evaporated Mg in the gas started to exceed the concentration of N$_2$ at a temperature of 1776 °C. The Mg partial pressure changed from the value equal to $5.89 \times 10^{-7}$ bar at 1000 °C to 0.72 bar at 1792 °C. At a temperature of 1756 °C, the concentration of Al$_2$O started to exceed Al as well. At temperatures above 1792 °C, an equilibrium could not be calculated, mainly because the activity of N$_2$ in the gas phase was too low to ensure the stability of AlN. The calculated activities of gas species are presented in Figure 3a. The partial pressure of oxygen increased with temperature from a value of $5.25 \times 10^{-27}$ bar at 1000 °C to a value of $3.02 \times 10^{-16}$ bar at 1792 °C. At partial pressures of O$_2$ higher than the equilibrium line, MgAl$_2$O$_4$ was stable and coexisted with MgO, whereas at partial pressures of O$_2$ below the equilibrium line, MgAl$_2$O$_4$ was not stable, decomposing with the formation of MgO and AlN phases due to a reaction of Al$_2$O$_3$ with nitrogen from the gas phase. The calculated temperature dependence of the oxygen partial pressure for an equilibrium of MgAl$_2$O$_4$ with MgO, AlN, and gas is presented in Figure 3b. This line indicates the lowest partial pressure boundary, where spinel is stable. It is shown as a dashed line in Figure 2b.

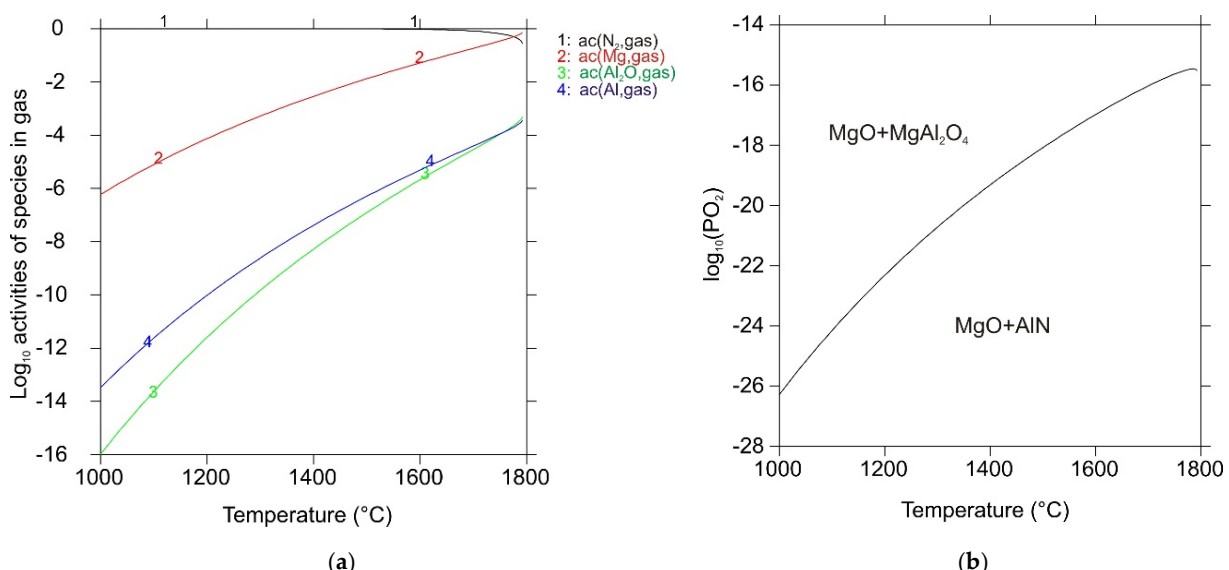

**Figure 3.** Equilibrium of MgAl$_2$O$_4$ with MgO and AlN in the presence of gas as a function of temperature: (**a**) activities of gas species and (**b**) partial pressure of oxygen (in bar).

Based on the obtained results, the conclusion can be drawn that MgAl$_2$O$_4$ is stable in its equilibrium with the gas phase at partial pressures below the phase boundary (solid line in Figure 2b), whereas an excess of Al$_2$O$_3$ reacts with N$_2$ from the gas phase, forming AlN. However, with a decrease in the oxygen partial pressure, spinel becomes unstable and decomposes, forming MgO and Al$_2$O$_3$. The latter reacts with nitrogen, forming AlN. Therefore, the line presented in Figure 3b represents the stability limit of spinel, which is also shown in Figure 2b. Additionally, the partial pressure of Mg is higher, if spinel coexists with MgO and AlN, and evaporation of Mg can occur during sintering.

Calculations in the Al$_2$O$_3$-AlN system were performed using the description of Dumitrescu and Sundman [38] for solid and liquid phases and the SSUB (SGTE) database for the gas phase (see Figure 4). They showed that Al$_2$O$_3$ and gas were in equilibrium with AlN at temperatures up to 1594 °C, where AlON spinel formed at an oxygen partial pressure equal to $1.07 \times 10^{-16}$ bar. At higher temperatures, AlON spinel could be in equilibrium either with Al$_2$O$_3$ and gas or AlN and gas. Phase R27 (an Al$_9$O$_3$N$_7$ polytype

with trigonal structure) appeared at 1883 °C. In equilibrium with AlON spinel, AlN, and gas, the oxygen partial pressure equaled $2.51 \times 10^{-14}$ bar. At temperatures above 1883 °C, the R27 phase could coexist with gas and either with AlON spinel or with AlN. At temperatures below 1531 °C, the major species in gas was $N_2$, followed by Al, $Al_2O$, and AlO. However, at higher temperatures, the sequence changed and the concentration of $Al_2O$ exceeded the Al concentration in the gas phase. The calculated activities of gas species are presented in Figure 4a. The calculated partial pressure of oxygen changed from a value equal to $1.23 \times 10^{-25}$ bar at 1000 °C for the equilibrium of $Al_2O_3+AlN+$gas to a value of $1.29 \times 10^{-11}$ bar for the equilibrium of AlON spinel with $Al_2O_3$ and gas at 2000 °C. AlON spinel was stable at oxygen partial pressures from $1.07 \times 10^{-16}$ bar at 1594 °C to a range of between $1.74 \times 10^{-13}$ bar and $1.29 \times 10^{-11}$ bar at 2000 °C. The results of these calculations are presented in Figure 4b.

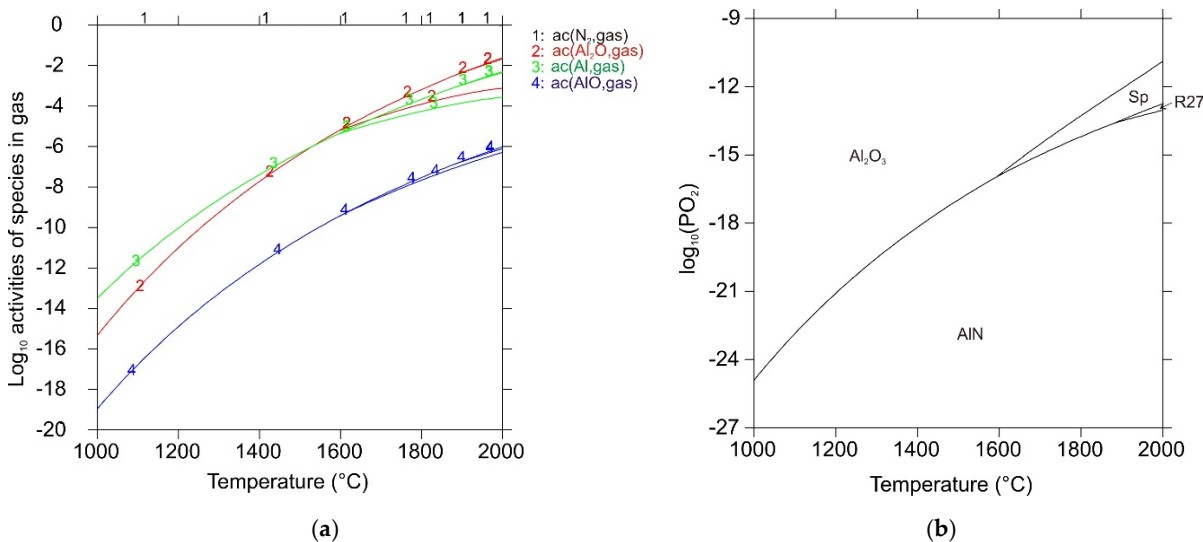

(a)  (b)

**Figure 4.** Phase equilibria in the $Al_2O_3$-AlN system $Al_2O_3+$AlN, $Al_2O_3+$spinel AlON, spinel AlON+AlN, R27+spinel AlON, and R27+AlN in the presence of gas: (**a**) temperature dependence of the activities of gas species and (**b**) temperature dependence of the partial pressure of oxygen (in bar).

AlON spinel formed at a temperature higher than 1594 °C and existed in limited ranges of oxygen partial pressures. The partial pressure of $Al_2O$ was higher than Al at the conditions of spinel stability, but the values were quite similar to each other, and a loss of both Al and $Al_2O$ can occur in conditions of sintering.

Furthermore, the density functional theory (DFT) [39,40] was applied to investigate reactions that could explain the presence of some of the major gas species presented in Figures 2a, 3a and 4a. These calculations were done using the Quantum ESPRESSO code [41–43] version 6.7. For all calculations, PAW [44] pseudopotentials taken from PSlibrary [45] version 1.0.0 were employed. All computations on the solids were performed using the PBEsol exchange-correlation functional [46], which is specifically tailored to solid state calculations. For the convergence tests, the structure optimization, and the vibrational analysis of the gaseous species, the PBE functional [47] was utilized. Nevertheless, the total energies of these gases were subsequently evaluated with PBEsol in order to enable comparisons to data from the solid phase. Before structure optimization, the kinetic energy cutoffs for the wave function and density as well as the k grid size for all investigated systems were chosen with a total energy convergence tolerance of 1 meV/atom. Moreover, all DFT results presented here were evaluated on optimized structures, with maximum atomic forces below 0.1 mRy/Bohr for solids and gases and, additionally, a maximum unit cell pressure below 0.1 kbar for solids. Within the quasi-harmonic approximation for the solid components and the ideal gas approximation for the gaseous components, the

standard Gibbs energies of the $\Delta_r G^\circ$ reaction were calculated for a temperature of 1500 °C and an external pressure of 1 bar, as shown in Table 2.

**Table 2.** The standard Gibbs energies of reaction $\Delta_r G^\circ$ at 1500 °C and 1 bar, calculated using DFT and compared to experimental references [48].

| Reaction No. | Chemical Equation | $\Delta_r G^\circ{}_{DFT}$ (kJ mol$^{-1}$) | $\Delta_r G^\circ{}_{ref}$ (kJ mol$^{-1}$) |
|:---:|:---:|:---:|:---:|
| 1 | $MgO_{(s)} \rightarrow Mg_{(g)} + {}^{1}/_{2}\, O_{2(g)}$ | 325.4 | 366.3 |
| 2 | $MgAl_2O_{4(s)} \rightarrow Al_2O_{3(s)} + Mg_{(g)} + {}^{1}/_{2}\, O_{2(g)}$ | 351.6 | 408.3 |
| 3 | $Al_2O_{3(s)} + 4\, AlN_{(s)} \rightarrow 3\, Al_2O_{(g)} + 2\, N_{2(g)}$ | 681.9 | 793.4 |
| 4 | $MgAl_2O_{4(s)} + 4\, AlN_{(s)} \rightarrow 3\, Al_2O_{(g)} + Mg_{(g)} + {}^{1}/_{2}\, O_{2(g)} + 2\, N_{2(g)}$ | 1033.4 | 1201.7 |
| 5 | $AlN_{(s)} \rightarrow Al_{(g)} + {}^{1}/_{2}\, N_{2(g)}$ | 247.0 | 230.8 |
| 6 | $2\, Al_{(g)} + {}^{1}/_{2}\, O_{2(g)} \rightarrow Al_2O_{(g)}$ | −529.3 | −485.5 |

According to Table 2, most reactions were predicted to be endergonic at 1500 °C and 1 bar by DFT, with the exception of reaction No. 6, a statement that is in agreement with experimental references [48]. Moreover, DFT managed to capture the trends of the experimental results, e.g., showing the same order of reactions, going from the most exergonic to the most endergonic. If we leave out reaction No. 4, which is the sum of reactions No. 2 and No. 3, we arrive at a mean absolute deviation of 53.8 kJ/mol for DFT when compared to experimental values. By contrasting the exergonic reaction No. 6 with the endergonic reactions No. 3 and No. 4, we can deduce that Al$_2$O is primarily formed from gaseous aluminum atoms, rather than as the product of solid-state reactions between Al$_2$O$_3$ and AlN or MgAl$_2$O$_4$ and AlN. The deviations between DFT and experimental references can be partially traced back to the approximations used for the calculation of the thermochemical data, i.e., the ideal gas approximation and the quasi-harmonic approximation. However, most of the error was expected to stem from the exchange-correlation functionals, which are responsible for inaccuracies in the calculated vibrational frequencies and, especially, total energies.

## 3. Methods and Experiments

### 3.1. Sample Preparation

As starting materials, Al$_2$O$_3$ (Martoxid$^\circledR$ MR70, Martinswerk GmbH, Bergheim, Germany, 99.8 wt%), MgO (Refratechnik Steel GmbH, Düsseldorf, Germany, 97.5 wt%), and AlN (Grade C, H. C. Starck, abcr GmbH, Karlsruhe, Germany, 97.9 wt%) were used. The true densities of the starting mixtures were measured using a helium pycnometer (Accupyc 1330, Micromeritics GmbH, Unterschleissheim, Germany) following the standard DIN 66137-2 (see Table 3).

Considering literature sources reviewed in the introduction (see Table 1) and the phase diagram proposed by Willems in 1992 [4], a simplex lattice design was applied in the most promising region of the phase diagram for a high conversion degree to MgAlON in the AlN-Al$_2$O$_3$-MgO pseudo-ternary system. The simplex design is shown in Figure 5, with the actual component space within the boundaries of 0.45–0.7 for $x_{Al_2O_3}$, 0.2–0.45 for $x_{MgO}$, and 0.1–0.35 for $x_{AlN}$.

Additionally, two compositions from within the component space were prepared and investigated, which were reported previously [23,49] to yield a high conversion degree at temperatures around 1500 °C. The experimental points and compositions of the simplex design, including the two additional points, are given in Figure 5 and Table 3.

**Table 3.** Compositions of the powder mixtures (mole fractions of AlN, MgO, and $Al_2O_3$) for sample manufacturing and their average densities $\rho_{mix}$ and particle sizes $d_{80}$.

| Mix No. | $x_{AlN}$ | $x_{MgO}$ | $x_{Al_2O_3}$ | $\rho_{mix}$ (Ø) (t m$^{-3}$) | $d_{80}$ (Ø) (µm) |
|---|---|---|---|---|---|
| 1 | 0.35 | 0.2 | 0.45 | 3.769 | 10.8 |
| 2 | 0.267 | 0.2 | 0.533 | 3.85 | 9.97 |
| 3 | 0.183 | 0.2 | 0.617 | 3.899 | 9.37 |
| 4 | 0.1 | 0.2 | 0.7 | 3.942 | 8.85 |
| 5 | 0.267 | 0.283 | 0.45 | 3.815 | 14.16 |
| 6 | 0.183 | 0.283 | 0.533 | 3.868 | 13.19 |
| 7 | 0.1 | 0.283 | 0.617 | 3.916 | 12.38 |
| 8 | 0.183 | 0.367 | 0.45 | 3.834 | 17.6 |
| 9 | 0.1 | 0.367 | 0.533 | 3.886 | 16.42 |
| 10 | 0.1 | 0.45 | 0.45 | 3.852 | 21.034 |
| 11 [48] | 0.27 | 0.22 | 0.51 | 3.841 | 10.91 |
| 12 [23] | 0.21 | 0.22 | 0.57 | 3.879 | 10.62 |

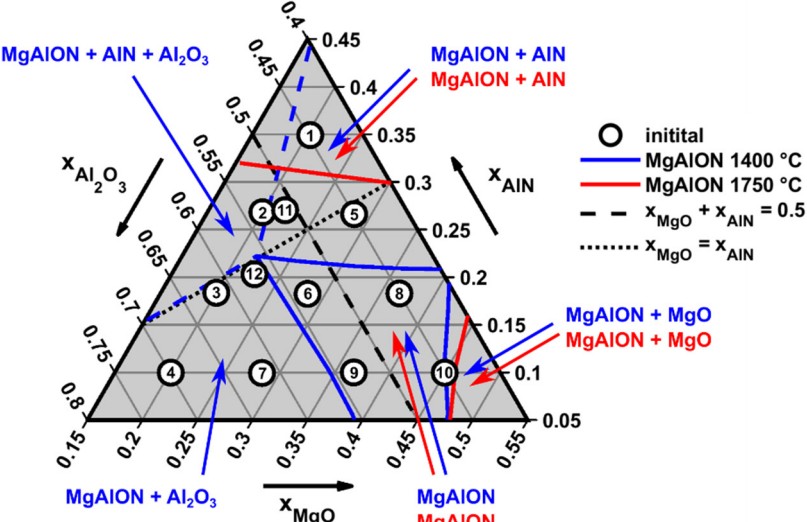

**Figure 5.** Compositions of the initial powder mixtures used, shown in a simplified phase diagram for the ternary MgO-AlN-$Al_2O_3$ system, according to Table 3. The homogeneity ranges of MgAlON and the adjacent multiphase regions for 1400 °C and 1750 °C were estimated according to the phase diagrams proposed by Willems [4].

To create the mixtures of $Al_2O_3$, MgO, and AlN powders needed to investigate the effect of the starting composition, dry ball mixing was chosen, as applied by Fruhstorfer et al. [50]. Considering the reactivity of AlN with aqueous grinding media, a dry mixing method was applied to avoid the reaction of AlN with water, which would form aluminum hydroxide and ammonia gas, causing a deficiency of nitrogen in the mixture [51,52].

For the preparation of dry ball-mixed samples, a tumbling mixer (Turbula System Schatz Type T2C, Willy A. Bachofen AG Maschinenfabrik Basel, Basel, Switzerland) with a three-dimensional oloid mixing motion at n = 30 rpm (=0.5 s$^{-1}$) was used. The powder mixtures were placed in a sealed container with a diameter D = 80 mm at a diameter–height ratio of 0.8 and a volume of approximately 0.5 L.

To calculate the needed size of the grinding balls, the following equation was applied for approximation, which is commonly used in the ball milling of ore or cement [53,54], due to the lack of specific data for tumble mixers:

$$d_{balls} = 0.24 \cdot (d_{80})^{1/2} \left( \frac{W_i \cdot \rho_{batch}}{n_{rel}} \right)^{1/3} D^{-1/6} \tag{1}$$

The relative revolution speed $n_{rel}$ was approximated accordingly from the relation for ball mills (Equation (2)) [54] and compared to the critical revolution speed $n_{crit}$ (Equation (3)). Effective stirring of the mill filling is only possible if the relative revolution speed is below the critical one, at which the contents of the mill would be centrifuged to the wall of the container [54].

$$n_{rel} = \pi \cdot n \cdot \sqrt{\frac{2 \cdot D}{g}} = \pi \cdot 0.5 \cdot \sqrt{\frac{2 \cdot 0.08}{g}} = 0.20 \tag{2}$$

$$n_{crit} = \sqrt{\frac{g}{2 \cdot \pi^2 \cdot D}} = \sqrt{\frac{9.81}{2 \cdot \pi^2 \cdot 0.08}} = 2.49 \tag{3}$$

Analogous to the experiments of Fruhstorfer et al. [50], the filling ratio of the mixing container was kept at approximately 33 vol%, following the optimal filling ratio of drum mixers, which is between 25 and 35 vol% [54]. Alumina grinding balls were chosen in order to avoid the introduction of impurities into the powder mixtures. The mixing time was chosen to be 10 min, as determined by Fruhstorfer at al., to deliver a homogeneous dry powder mixture in the tumbling mixer [50]. In the case of dry mixtures, long mixing times tend to promote segregation of powders with different grain sizes, rather than the formation of a homogeneous mixture [50]. After the mixing step, the powder was separated from the grinding balls by manually screening it through a 1.5 mm sieve. To shape the samples, a manually operated torque press was used to press the powder into tablets with a weight of 3 g each, a diameter of 15 mm, and a height of 8 mm at 30 MPa.

### 3.2. Reaction Sintering

The prepared sample tablets were placed in open alumina crucibles, which in turn were kept inside porous graphite containers within an electric high-temperature furnace (HT-1600-GT-Vac, Linn High Therm GmbH, Hirschbach, Germany). The graphite containers were used as oxygen getters to reactively lower any residual oxygen content present in the furnace atmosphere, positively influencing the synthesis process and hindering oxidation of remaining AlN or formed MgAlON (cf. Figure 4 and [4]).

Before the thermal treatment was started, the furnace was evacuated for one hour and subsequently flushed with nitrogen (99.99%). After the furnace chamber was fully filled with nitrogen, the constant nitrogen flow was adjusted to 3.5 L min$^{-1}$. This flow rate enabled the maintenance of a slight nitrogen overpressure inside the furnace chamber ($1.1 \text{ bar} \leq p \leq 1.3 \text{ bar}$), which prevented air leakage into the furnace, maintaining the protective, non-oxidizing atmosphere. A pressure equalization with the air leak of the furnace was determined to occur at a nitrogen flow rate of 3.2 L min$^{-1}$ [54]. The heating rate was set to 5 K min$^{-1}$, and the two batches of samples were held at a sintering temperature of 1500 °C for 3 or 6 h.

The synthesis parameters were in line with the previously performed synthesis of $Si_2N_2O$ by Fruhstorfer et al. [55]. After the holding time ended, the samples were cooled down in the furnace at a rate of 5 K min$^{-1}$.

### 3.3. X-ray Diffraction and Chemical Analysis

Sintered tablets of each composition and heat treatment route were ground using a Retsch RS200 tungsten carbide vibratory disk mill for 2 min at 700 rpm. The weight fractions of the formed spinel phase and the unconsumed initial phases ($Al_2O_3$, MgO, AlN), as well as their respective lattice parameters, were determined ex situ by X-ray diffraction at ambient conditions.

The measurements were performed in Bragg–Brentano geometry on an X'Pert PRO (Malvern Panalytical, Malvern, United Kingdom) utilizing CuK$\alpha$ radiation ($\lambda_{K\alpha 1} = 0.15406$ nm, $\lambda_{K\alpha 2} = 0.15444$ nm). The equatorial divergence of the primary beam was limited to 0.5° and the axial divergence was restricted by Soller collimators (2.3°) both in the primary and in the secondary beam path. The diffraction patterns were recorded in the range of 20°–90° with

an effective step size of 0.013° and an effective measurement time of 29 s (Pixcel1D detector, Malvern Panalytical, Malvern, UK).

For the processing of the recorded data, the Rietveld routine TOPAS (AXS Bruker, Karlsruhe, Germany, Version 5) was used. The crystal structures of individual phases (see Table 4) were taken from the Inorganic Crystal Structure Database (ICSD) [56]. For the assessment of the quality of the refinement, the weighted profile R-factor ($R_{wp}$) was utilized (Equation (4)):

$$R_{wp} = \sqrt{\frac{\sum_i w_i \left(y_{o,i} - y_{c,i}\right)^2}{\sum_i w_i y_{o,i}^2}} \cdot 100\% \text{ with } w_i = \frac{1}{\sigma^2\left(y_{o,i}\right)} = \frac{1}{y_{o,i}} \tag{4}$$

In Equation (4), $y_{o,i}$ and $y_{c,i}$ are the observed and the calculated intensities at the ith measurement point, respectively. The standard uncertainty $\sigma\left(y_{o,i}\right)$ of each measurement point for pulse counting detectors was applied [57].

**Table 4.** Crystal structure data used for the Rietveld refinement of $Al_2O_3$, MgO, AlN, and $MgAl_2O_4$.

| Phase | Space Group | ICSD Nr. | Original Reference |
|---|---|---|---|
| $Al_2O_3$ | $R\bar{3}c$ | 160604 | [58] |
| MgO | $Fm\bar{3}m$ | 52026 | [59] |
| AlN | $P6_3mc$ | 34475 | [60] |
| $MgAl_2O_4$ | $Fd\bar{3}m$ | 31373 | [61] |

For the Rietveld refinement of the oxynitride spinel phase (MgAlON), the structure of $MgAl_2O_4$ was employed (see Table 4). This approach caused an error in the refined weight fraction that was far below the typical error of the quantitative phase analysis. Since Al and Mg as well as O and N are neighboring elements, their atomic scattering factors are nearly identical [62]. Consequently, variations in the [Mg]/[Al] and [N]/[O] ratios did not change the intensities of the diffraction lines. Furthermore, the weight per unit cell ($Fd\bar{3}m$), assuming a constant anion sublattice [27], varied between 1087.6 u, 1138.1 u, and 1143.6 u for $\gamma$-$Al_2O_3$ (or $Al_{2.6667}O_4$), $MgAl_2O_4$, and $Al_3O_3N$, respectively. Thus, the bulk densities, which are required for conversion of the volume fractions obtained directly from the XRD intensities into weight fractions, were in the range of 3.55 g cm$^{-3}$ $\leq \rho \leq$ 3.75 g cm$^{-3}$ for $x_{Al_2O_3} \leq 0.5$ [4,23]. Finally, the linear attenuation coefficients of Al-O-based spinels, which may also affect the diffracted intensities, were in the range of 105 cm$^{-1}$ < $\mu$ < 120 cm$^{-1}$ for CuK$\alpha_1$ radiation [63].

For the determination of the phase fractions beside the scale factor, the lattice parameters, and an overall isotropic temperature factor, no further structural parameters of the phases (see Table 4) were refined. The only exception were the cation vacancies for the MgAlON phase, which were adjusted to accommodate slight intensity variations for different compositions. The impact on the refined weight fraction with a maximum deviation of around 1 wt% was marginal and within the common error range of the Rietveld refinement and did not affect the observed results and trends. This is exemplarily shown for the sample with the highest $Al_2O_3$ content (sample No. 4, Figure S5c), where MgAlON was expected to exhibit the largest amount of cation vacancies (Section 2). The amount of cation vacancies of 5.1% for this sample was not exceeded by any other sample. In accordance with Zong et al. [64] these variations primarily affected the intensity of the diffraction line 111 (2$\theta \approx 18°$). However, at low diffraction angles for powder samples these intensities were also influenced by microabsorption and sample roughness. Thus, a discussion of these sensitive parameters can only be conducted in combination with complementary investigations including a full chemical analysis (not only of the nitrogen content) and nuclear magnetic resonance spectroscopy, as demonstrated by Zong et al. [64]. Such a detailed evaluation is far beyond the scope of this work, and the presented refinement procedure is sufficient for the determination of the weight fractions and the mean lattice

parameter of the MgAlON phase. In selected samples, the overall nitrogen content was analyzed using carrier gas extraction in order to be able to estimate the nitrogen content in MgAlON from the phase fractions of MgAlON, AlN, $Al_2O_3$, and MgO determined by XRD. The concentration of nitrogen was determined using carrier gas extraction (ONH 836, Leco, St. Joseph, USA) on 7.5 mg of sintered, disk-milled powder of each selected sample. Four repeated measurements were performed for each sample; graphite crucibles were used along with tin and nickel as flux media.

### 3.4. Raman Spectroscopic Measurements

Six of the samples, i.e., the samples with mix No. 1, 7, and 12 (sintered for 3 and 6 h), were additionally investigated with Raman spectroscopy. The measurements were performed with a UV Labram HR800 Horiba Jobin equipped with a 2400 cm$^{-1}$ grating and a Peltier-cooled CCD detector. For excitation, the 325 nm line of an He:Cd laser was used. The power of the laser light at the sample surface was approximately 2.8 mW. The light was focused by a $\times 40$ objective. For each sample, three measurements were done. The recorded spectra were baseline corrected with the adaptive iteratively reweighted penalized least squares method proposed by Zhang et al. [65].

### 3.5. Sessile Drop Tests on Selected Sintered Samples

Sessile drop tests were conducted on selected sintered ceramic samples to evaluate their wettability, interfacial reactivity, and therefore, the applicability of the material in light metal melt filtration. Following the technique described by Fankhänel et al. [66], freshly cut AlSi7Mg samples ($60 \pm 2$ mg, Trimet Aluminum AG) were placed on the respective as-fired ceramic surfaces. The ceramic substrates were not polished so as to approximate the unpolished surface of a ceramic foam filter. Before the sessile drop test, their mean surface roughness was determined using an optical confocal microscope (μsurf explorer, NanoFocus AG, Oberhausen, Germany) in accordance with DIN EN ISO 25178, measuring the surface roughness of each substrate in 4 places.

The ceramic and AlSi7Mg samples were placed in a high-temperature furnace equipped with a high vacuum and inert gas system (Gero GmbH, Forchheim, Germany). A digital image analyzer was used to carry out contact angle measurements between the ceramic substrate and molten aluminum droplet for 180 min at 950 °C. Contact angles to the left and right side of the metal droplet were evaluated. The heating rate of the furnace was set to 350 °C h$^{-1}$ and its chamber had been evacuated, reaching pressures of $2.2 \times 10^{-5}$ mbar during holding and measuring time.

As described by Eustatopoulos et al. [67], the contact angles ($\theta_{cal}$) were calculated using the following Equation (5). The AlSi7Mg droplets used in the experiment were lighter than 100 mg; therefore, their shape can be assumed to have been a spherical segment with a measured height (h) and diameter (d):

$$\theta_{cal} = 2\arctan\left(2hd^{-1}\right) \tag{5}$$

## 4. Experimental Results and Discussion

### 4.1. Influence of Starting Composition and Sintering Duration on the Conversion Degree

XRD analysis shows that the weight percentage of MgAlON formed in sintered samples increased with increasing fractions of MgO in the starting mixture for small amounts of AlN (approx. $x_{AlN} < 0.15$) and with sintering time, as can be seen in Table 5 and Figure 6a,e. Respective XRD data are presented in Supplementary Figures S1–S13. After 3 h of sintering, the average weight percentage of MgAlON was approximately 76 wt% and increased to 92 wt% for a holding time of 6 h. From the response surfaces (see Figure 6a,e), a conclusion can be drawn that the reaction state did not come close to thermodynamic equilibrium after 3 h. After a dwell time of 6 h, the system seemed notably closer to equilibrium and the area enclosed by the 95 wt% contour line of MgAlON

(see Figure 6e) coincided well with the homogeneity region reported by Willems around 1400 °C (cf. Figure 6h) [4].

**Table 5.** Conversion degree (wt% of MgAlON spinel determined using XRD) of powder mixtures to MgAlON spinel depending on their starting composition and holding time. See Supplementary Table S1 for the respective amounts of residue phases $Al_2O_3$, AlN, and MgO after sintering.

| Mix No. | Starting Composition (Mole Fraction) (x) | | | Conversion Degree to MgAlON Spinel (wt%) after: | |
|---|---|---|---|---|---|
| | AlN | MgO | $Al_2O_3$ | 3 h of Sintering | 6 h of Sintering |
| 1 | 0.35 | 0.2 | 0.45 | 67 | 86 |
| 2 | 0.267 | 0.2 | 0.533 | 65 | 92 |
| 3 | 0.183 | 0.2 | 0.617 | 65 | 92 |
| 4 | 0.1 | 0.2 | 0.7 | 59 | 76 |
| 5 | 0.267 | 0.283 | 0.45 | 82 | 89 |
| 6 | 0.183 | 0.283 | 0.533 | 82 | 96 |
| 7 | 0.1 | 0.283 | 0.617 | 71 | 90 |
| 8 | 0.183 | 0.367 | 0.45 | 93 | 94 |
| 9 | 0.1 | 0.367 | 0.533 | 93 | 99 |
| 10 | 0.1 | 0.45 | 0.45 | 97 | 99 |
| 11 | 0.27 | 0.22 | 0.51 | 70 | 92 |
| 12 | 0.21 | 0.22 | 0.57 | 68 | 96 |
| **Average conversion degree:** | | | | 76 | 92 |

The observed time- and composition-dependent changes in the weight percentages of MgAlON, MgO, AlN, and $Al_2O_3$ shown in Figure 6 can be correlated to general reactions during the sintering process, as the observed changes are mainly consistent with the reaction mechanism postulated by Yan et al. [68] for the synthesis of MgAlON via aluminothermic reduction. An initial formation of AlN from metallic Al was not required in our case and thus could not artificially delay the dissolution of AlN. MgO is therefor consumed first, independent of the starting composition (cf. Figure 6d), as it reacts with $Al_2O_3$, forming $MgAl_2O_4$:

$$MgO + Al_2O_3 \rightarrow MgAl_2O_4 \tag{6}$$

This reaction starts at temperatures of around 950 to 1150 °C [19,23,68,69]. Accordingly, Figure 6d shows that there were negligible amounts of MgO remaining after a sintering time of 3 h, whereas the amount of MgO remaining after 6 h was practically zero (see Table S1). This observation indicates that the formation of $MgAl_2O_4$ consumes almost any MgO present in the starting mixture. The $Al_2O_3$/AlN ratio of the mixture plays a secondary role for the conversion degree [23,70]. With increasing temperatures, the homogeneity range of magnesium aluminate spinel expands [30,31], facilitating the dissolution of alumina into the spinel, forming alumina-rich spinel containing cation vacancies (see Equation (7), with cation vacancies marked as $\square$, and n and x* being stoichiometric coefficients).

$$MgAl_2O_4 + n\ Al_2O_3 \rightarrow (1 + \tfrac{3}{4}\ n)\ Mg_{x^*}Al_{\frac{8-2x^*}{3}}\square_{\frac{1-x^*}{3}}O_4 \tag{7}$$

$$x^* = (1 + \tfrac{3}{4}\ n)^{-1}$$

The observations displayed in Figure 6c,g confirm an increasing consumption of $Al_2O_3$ at the longer sintering time of 6 h. Higher amounts of excessive $Al_2O_3$ in the mixture resulted in a longer dissolution process of $Al_2O_3$, resulting in a higher amount of cation vacancies in the spinel structure, as described in the preliminary thermodynamic considerations (Section 2) [32].

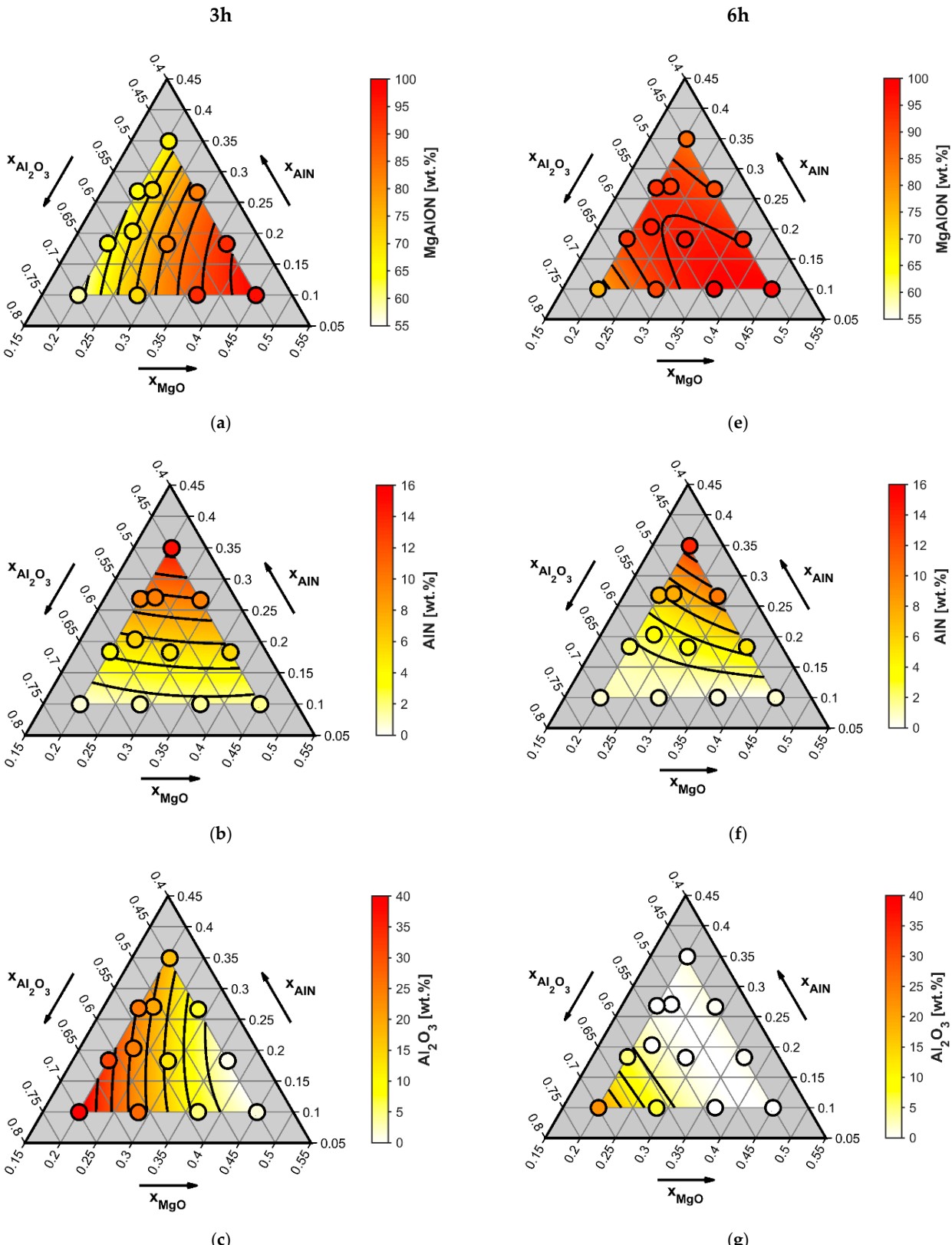

**Figure 6.** *Cont*.

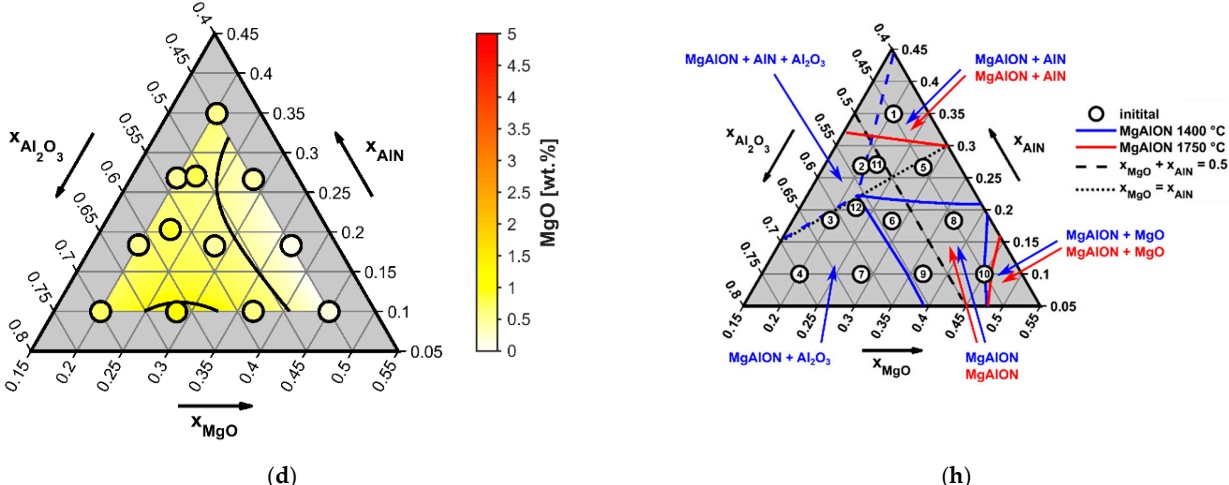

(d)          (h)

**Figure 6.** Weight percentages of formed and remaining initial phases as a function of initial composition within the quasi-ternary MgO-AlN-Al$_2$O$_3$ system after 3 h (a–d) and 6 h (e–g) of sintering at 1500 °C in nitrogen atmosphere. The weight percentages were obtained from XRD analysis (see Supplementary Table S1). The solid black contour lines mark the constant weight percentages of the respective compound. The weight percentages of MgO are not presented for 6 h, as its content had already decreased to nearly 0 wt% for all samples after 3 h of sintering. Panel (h) shows the positions of the samples from Tables 3 and 5 in the composition diagram along with the respective lines of the phase diagram proposed by Willems [4] for direct comparison.

The dissolution of AlN into the spinel structure is kinetically hindered to a larger extent [23], though it can be seen in Figure 6f that the amount of residual AlN was slightly lower on the alumina-rich side. This suggests an improved dissolution of AlN into spinel structures containing higher amounts of cation vacancies. This dissolution process leads to a decrease in the cation vacancy concentration within the spinel (see Equation (8), with x, x*, n*, and y being respective stoichiometric coefficients), whereas the amounts of Al$_2$O$_3$ and AlN that can be dissolved into the spinel structure depend on the homogeneity range of Mg$_x$Al$_{(8-2x-y)/3}$O$_{4-y}$N$_y$ at the sintering temperature [7].

$$\mathrm{Mg}_{x^*}\mathrm{Al}_{\frac{8-2x^*}{3}}\square_{\frac{1-x^*}{3}}\mathrm{O}_4 + n^*\mathrm{AlN} \rightarrow \left(1 + \frac{1}{4}\,n^*\right)\mathrm{Mg}_x\mathrm{Al}_{\frac{8-2x+y}{3}}\square_{\frac{1-x-y}{3}}\mathrm{O}_{4-y}\mathrm{N}_y \tag{8}$$

$$y = n^*\left(1 + \frac{1}{4}\,n^*\right)^{-1}, \quad x = x^*\left(1 + \frac{1}{4}\,n^*\right)^{-1}$$

The theoretical amount $n_0^*$ of AlN needed for the annihilation of all cation vacancies is

$$n_0^* = \frac{4}{3}\,(1 - x^*) \tag{9}$$

As the dissolution of Al$_2$O$_3$ produces structural vacancies in Mg$_x$Al$_{(8-2x-y)/3}$O$_{4-y}$N$_y$ that are annihilated by the dissolution of AlN, it can be assumed that AlN and Al$_2$O$_3$ dissolve simultaneously. Still, these dissolution processes compete and the dissolution of AlN is kinetically hindered compared to the dissolution of Al$_2$O$_3$. When the solution limit of AlN is reached ($\neq n_0^*$), an increase in the cation vacancy concentration by further dissolution of Al$_2$O$_3$ can promote a continued dissolution of AlN and vice versa. However, if the initial amount of AlN exceeds a certain limit, the AlN + MgAlON dual-phase region is entered (see Figure 6h). Its boundary can be estimated from Figure 6e to be around $x_{AlN} \approx 0.15$ at the sintering temperature of 1500 °C, as above, significant amounts of remaining AlN are observed. This value is slightly below the proposed boundary of Willems ($x_{AlN} \approx 0.20$) (Figure 6h) [4] at 1400 °C. Higher dwell times, as utilized in [4], may

lead to further dissolution of minor AlN amounts but also to its oxidation or evaporation, as indicated by the lower measured nitrogen contents compared to the expected contents (cf. Table 6 and Section 2). Therefore, we consider 0.15 to be more realistic than 0.2 for our synthesis conditions. Considering this, a conclusion can be drawn that higher weight fractions of MgAlON can be obtained if the amount of AlN in the initial mixture is kept below 0.15.

**Table 6.** Overall nitrogen contents of MgAlON-containing samples No. 4 and No. 9 sintered for 6 h, as determined via carrier gas hot extraction (CGHE), as expected from the initial composition (cf. Tables 3 and 5) and their ratio. The nitrogen contents in the remaining AlN and in MgAlON were calculated from the phase fractions obtained by XRD and from the overall nitrogen content. The absolute nitrogen content in MgAlON is related to the entire sample weight, including remnants of $Al_2O_3$ and MgO, whereas the relative content represents the weight fraction of nitrogen within the MgAlON phase.

| Substrate | Overall Nitrogen Content (wt%) | Nitrogen Content, Expected (wt%) | Ratio (Measured/ Expected) | Nitrogen Content in Remaining AlN (wt%) | Absolute Nitrogen Content in MgAlON (wt%) | Relative Nitrogen Content in MgAlON (w/w%) |
|---|---|---|---|---|---|---|
| MgAlON-containing sample No. 4 | 1.48 ± 0.17 | 1.67 | 0.89 ± 0.11 | 0.31 | 1.17 ± 0.17 | 1.5 ± 0.3 |
| MgAlON-containing sample No. 9 | 1.64 ± 0.15 | 1.91 | 0.86 ± 0.08 | 0.17 | 1.47 ± 0.15 | 1.5 ± 0.2 |

The noticeable decrease in AlN and $Al_2O_3$ after 6 h of sintering time (cf. Figure 6c,b,f,g) indicates the dissolution of both phases and their transformation into the spinel structure.

The remaining amounts of $Al_2O_3$ for sample Nos. 3, 4, and 7, as well as the steep slope within the response surface (cf. Figure 6f), suggest that the phase boundary between MgAlON and MgAlON + $Al_2O_3$ was shifted to slightly higher amounts of $Al_2O_3$ (approx. $0.65 \geq x_{Al_2O_3} \geq 0.6$) at 1500 °C, compared to the boundaries at 1400 °C, which were reported by Willems (Figure 6h) [4]. Considering the applied temperature of 1500 °C for the present sintering process, this is in good agreement.

At the sintering temperature of 1500 °C, a direct formation of γ-AlON from AlN and $Al_2O_3$ was not expected, as γ-AlON is not thermodynamically stable at temperatures below 1594 °C (cf. Section 2 and [37]). Nonetheless, it is possible that minor amounts of MgAlON may have been formed directly from the vapor phase due to secondary reactions involving gaseous species [23,68,71], which result from the low oxygen partial pressures and the concurrent presence of MgO and AlN at the beginning of the sintering process.

The highest conversion degree was reached for sample Nos. 9 and 10, reaching approximately 99 wt% of MgAlON for both (see Table 5). Considering Figure 6f and the negligibly low content of approximately $w_{AlN} = 0.6$ remaining in sample Nos. 9 and 10, it can be concluded that most of the nitrogen was dissolved in the MgAlON spinel of the sintered ceramic material.

As reported by Willems [68] and Granon et al. [23], the lattice parameter of the $Mg_xAl_{(8-2x-y)/3}O_{4-y}N_y$ spinel depends on the molar fractions of MgO and AlN ($x_{MgO}$ and $x_{AlN}$) dissolved within the spinel (not to be confused with the initial composition):

$$a \text{ [Å]} = 7.9 + 0.375\, x_{MgO} + 0.15\, x_{AlN} \tag{10}$$

When MgAlON is considered a pseudo-solid solution of MgO and AlN, the molar fractions of MgO and AlN can be related to the stoichiometric coefficients of $Mg_xAl_{(8-2x-y)/3}O_{4-y}N_y$ (x and y):

$$x_{MgO} = \frac{3x}{2(2+x+y)} \tag{11}$$

$$x_{AlN} = \frac{3y}{2(2+x+y)} \tag{12}$$

The mean lattice parameters determined via XRD analysis for the sintered MgAlON samples are summarized in Figure 7. For constant molar fractions of MgO in the starting composition, a slight increase in the lattice parameter could be seen for increasing AlN contents. This observation is in agreement with the lattice parameter dependence (Equation (10)) shown by Willems [70] and Granon et al. [23], and indicates the dissolution of AlN into the spinel phase.

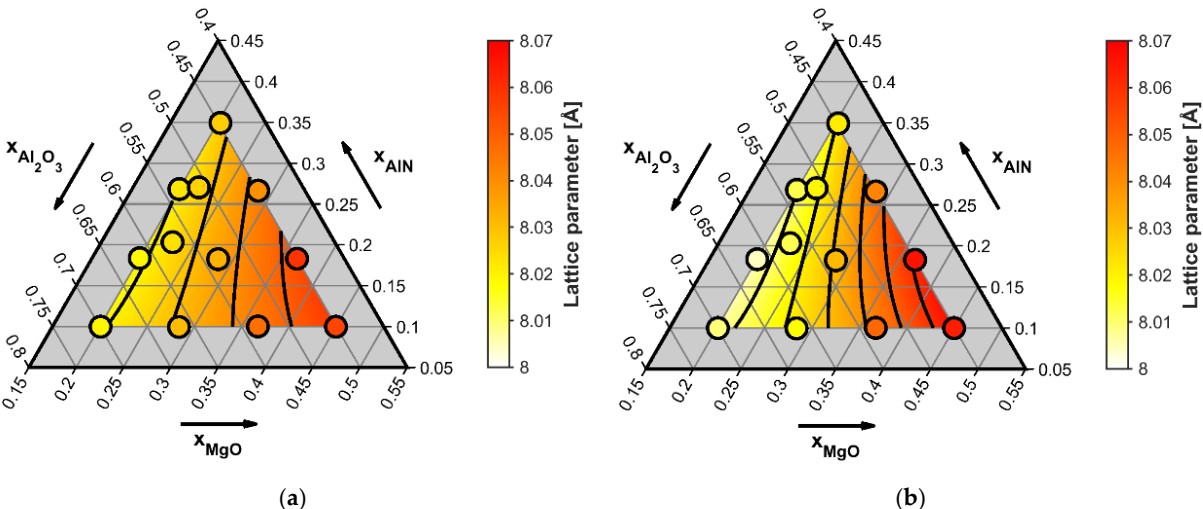

(a)　　　　　　　　　　　　　　　　(b)

**Figure 7.** Mean lattice parameters of MgAlON as a function of the initial composition within the quasi-ternary MgO-AlN-Al$_2$O$_3$ system after 3 h (**a**) and 6 h (**b**) of sintering time at 1500 °C in nitrogen atmosphere as a function of the initial composition (cf. Supplementary Table S2). The black contour lines correspond to constant lattice parameters.

Considering the remaining amounts of Al$_2$O$_3$ and AlN after 3 and 6 h of reactive sintering (cf. Figure 6b,c,g,f) and the generally small lattice parameters observed for aluminum-rich spinel [23], the decrease in lattice parameters for samples with $x_{MgO} < 0.3$ reflects the ongoing dissolution of Al$_2$O$_3$ into the spinel structure (see Equation (7)).

Furthermore, the evaluation of lattice parameters helps to draw a conclusion on the impact of magnesium loss on the formed spinel structure [19,21,23].

The lattice parameters were reported to be very sensitive to the magnesium content of the spinel [23]. There was only a slight increase in the lattice parameters of sample Nos. 9 and 10 in the MgO-rich corner of the simplified phase diagram MgO-AlN-Al$_2$O$_3$, as the sintering time increased from 3 to 6 h (see Figure 7), with MgO and Al$_2$O$_3$ having been largely consumed in this region after 3 h (see Figure 6c,d). Therefore, the loss of magnesium due to evaporation or parasitic reactions can be seen as insignificant after the spinel formation. Otherwise, the lattice parameters would have shown a notable decrease. This corresponds with the findings of the preliminary thermodynamic calculations (see Section 2) stating that the evaporation of magnesium is only largely facilitated if MgO coexists with the spinel phase and is not consumed.

### 4.2. Raman Measurements

A Raman spectrum of sample No. 7 (6 h of sintering) is shown in Figure 8. It exhibited four peaks around 315, 410, 685, and 795 cm$^{-1}$. All of these peaks also occurred in spectra for the spinel MgAl$_2$O$_4$, where they were assigned to the T$_{2g}$(1), E$_g$, T$_{2g}$(3), and A$_{1g}$ modes [72,73]. Of special interest is the E$_g$ mode at 410 cm$^{-1}$, since a broadening of that peak is related to the cation disorder in the spinel structure [74,75]. It should be noted that the E$_g$ peak exhibited a shoulder peak around 380 cm$^{-1}$, occurring only for disorder systems [75], but for the present purpose, it is reasonable to treat it as one peak.

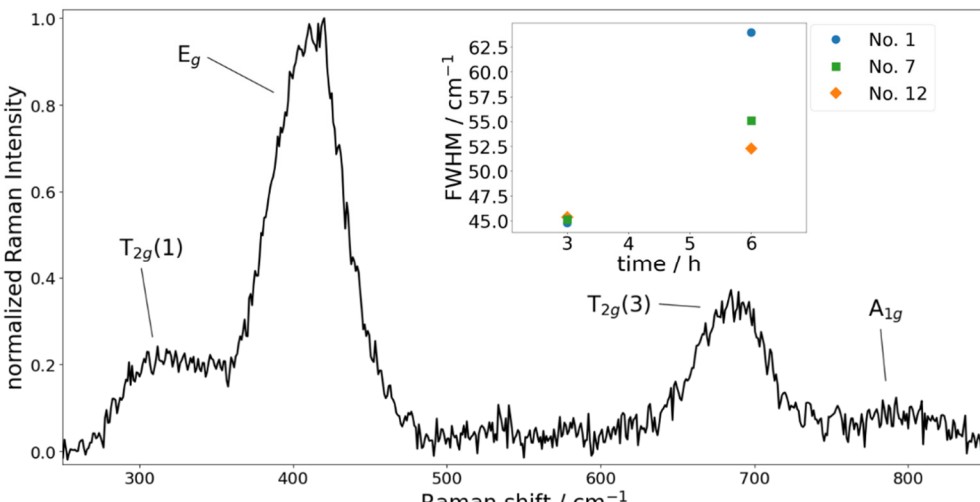

**Figure 8.** Exemplary Raman spectrum of the sample with mix No. 7 and a sintering time of 6 h. The four visible peaks are assigned to modes occurring in the spinel $MgAl_2O_4$. The inset shows how the peak width of the $E_g$ peak at 410 cm$^{-1}$ increases with sintering time.

In order to obtain the peak width (FWHM—full width at half maximum) of the $E_g$ mode, this peak and the overlapping $T_{2g}(1)$ peak around 315 cm$^{-1}$ were fitted with Gaussians. The obtained FWHM of the $E_g$ peaks are shown in the inset of Figure 8. All investigated samples showed greater FWHM for the samples with a sintering time of 6 h compared to the samples sintered for 3 h, indicating an increase in the cation disorder of the spinel structure, which was caused by the nitrogen included in the spinel structure. Thus, the Raman results confirm the findings of the XRD measurements and thermodynamic considerations, namely, that the dissolution of AlN and $Al_2O_3$ in the spinel structure after a dwell time of 6 h was well advanced. This, in turn, led to a higher amount of tetrahedrally coordinated Al, which is associated with an increase in cation disorder and thus, a broadening of the $E_g$ peak.

*4.3. Nitrogen Content of Selected Samples*

As no direct information about the nitrogen content in MgAlON could be obtained from the XRD analysis, two sintered samples were selected for the measurement of the nitrogen content by means of carrier gas extraction. The combination of these experimental techniques was sufficient, since AlN and MgAlON were the only phases containing nitrogen. $Al_2O_3$ and MgO were expected to be nitrogen-free. The first selected sample type (starting composition No. 9, see Table 5) showed the highest conversion degree to MgAlON after 6 h of reactive sintering at 99 wt%; the second sample type (starting composition No. 4, see Table 5) exhibited the lowest conversion degree to MgAlON of all tested samples after 6 h of sintering, reaching only 76 wt%.

The measured nitrogen contents of the samples are presented in Table 6.

Considering the amount of AlN in samples No. 4 (0.9 wt%) and No. 9 (0.5 wt%) detected by XRD after reactive sintering for 6 h (cf. Figure 6f) and the relative weight fraction of nitrogen in AlN ($w_N^{AlN} = 0.34$), the relative nitrogen content in MgAlON was determined as $w_N^{MgAlON} = \frac{w_N - w_{AlN} \, w_N^{AlN}}{w_{MgAlON}} \approx 0.015$. The same values for both spinels are consistent with the identical amount of AlN in the initial powder mixtures (mole fraction of 0.10, see Tables 3 and 5). The measured nitrogen contents were slightly lower than the values expected according to the initial composition, which indicated a slight oxidation of the samples or the formation of minor amounts of volatile $Mg_3N_2$ [23].

### 4.4. Wettability of Selected Sintered Samples by AlSi7Mg

Sample Nos. 4 and 9, sintered for 6 h and exhibiting the lowest (76 wt%) and highest (99 wt%) respective conversion degrees to MgAlON (cf. Table 5), were selected for the sessile drop tests using the aluminum alloy AlSi7Mg.

Table 7 shows the mean roughness parameters $S_a$, which were determined for the as-sintered surfaces, comparing them to the roughness parameters determined by Fankhänel et al. for as-fired $Al_2O_3$- and $MgAl_2O_4$-coated substrates, which are state-of-the-art filter ceramics that can be seen as benchmark materials [66]. The measured surface roughness parameters $S_a$ of the MgAlON-containing surfaces synthesized in this work were seen to be in a similar range to the one measured for the $MgAl_2O_4$ spinel sample of Fankhänel et al. [66], facilitating the comparability of the sessile drop experiments, with the influence of the surface roughness being low.

**Table 7.** Measured mean surface roughness parameters $S_a$ for MgAlON- and $Al_2O_3$-/$MgAl_2O_4$-containing [66] substrates.

| Substrate | Ø $S_a$ (μm) |
|---|---|
| As-sintered MgAlON-containing sample, composition No. 4 I | $4.2 \pm 1.7$ |
| As-sintered MgAlON-containing sample, composition No. 4 II | $4.8 \pm 2.2$ |
| As-sintered MgAlON-containing sample, composition No. 9 I | $4.0 \pm 0.5$ |
| As-sintered MgAlON-containing sample, composition No. 9 II | $3.8 \pm 1.0$ |
| As-fired alumina substrate, $MgAl_2O_4$ coating [66] | $3.6 \pm 0.1$ |
| As-fired alumina substrate, $Al_2O_3$ coating [66] | $1.7 \pm 0.2$ |

The wettability of two samples of each type was tested, and the respective measurements corresponded well with each other, showing the reproducibility of the results, as shown in Figure 9. The average standard deviation of the right and left contact angles never exceeded 0.6.

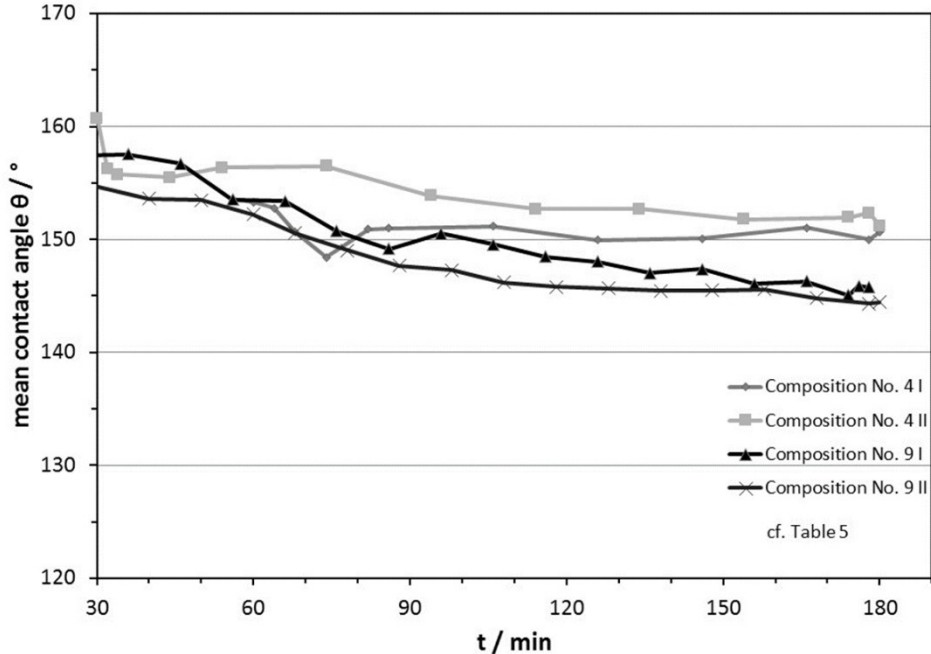

**Figure 9.** Measured time dependence of mean contact angles θ of AlSi7Mg droplets on MgAlON-containing substrates.

The contact angles measured between the MgAlON-containing substrate and the left and right side of the AlSi7Mg droplet after 90 and 180 min of contact time are listed in Table 8. The average contact angles reported by Fankhänel et al. for $Al_2O_3$ and $MgAl_2O_4$ surfaces [66] are listed as benchmarks, being state-of-the-art ceramic foam filter materials.

**Table 8.** Measured contact angle θ at 950 °C for MgAlON- and $Al_2O_3$-/$MgAl_2O_4$-containing [66] substrates.

| Substrate | 90 Min of Contact Time | | | 180 Min of Contact Time | | | Overall Ø Contact Angle |
|---|---|---|---|---|---|---|---|
| | Left | Right | Ø | Left | Right | Ø | |
| As-sintered MgAlON-containing sample, composition No. 4 I | 150.9° | 151.0° | 150.9° | 150.7° | 150.7° | 150.7° | 150 ± 0.5° |
| As-sintered MgAlON-containing sample, composition No. 4 II | 153.9° | 153.9° | 153.9° | 151.2° | 151.2° | 151.2° | 152 ± 0.5° |
| As-sintered MgAlON-containing sample, composition No. 9 I | 149.2° | 149.2° | 149.2° | 145.7° | 145.6° | 145.6° | 146 ± 0.5° |
| As-sintered MgAlON-containing sample, composition No. 9 II | 147.7° | 147.7° | 147.7° | 144.4° | 144.4° | 144.4° | 145 ± 0.6° |
| As-fired alumina substrate, $MgAl_2O_4$ coating [66] | | | 129 ± 1° | | | 122 ± 1° | |
| As-fired alumina substrate, $Al_2O_3$ coating [66] | | | 103 ± 1° | | | 100 ± 1° | |

Fankhänel et al. reported $MgAl_2O_4$ to be non-wettable by the AlSi7Mg alloy, with the contact angle not sinking below 120° during the holding time of 3 h at 950 °C (see Table 8), with traces of oxygen and aluminum being detected at the interface via EDX analysis, indicating an interfacial reaction [66]. According to Eustatopoulos et al. [67], wetting angles above 120° signify non-wetting as well as non-reactive behavior for polycrystalline ceramic materials in contact with metal melts. Similar traces of interfacial reactions were reported by Fankhänel et al. for the $Al_2O_3$-coated substrate, additionally showing contact angles around 100°, implying the wettability of the substrate by the AlSi7Mg melt, while also considering the lower roughness parameter of the substrate, facilitating higher wettability (see Tables 7 and 8) [66].

Compared to this benchmark, both of the MgAlON-containing samples, manufactured by means of reaction sintering in this work, showed contact angles that were remarkably higher than those of the commonly used $MgAl_2O_4$ and $Al_2O_3$ coatings. As can be seen in Figure 9, the mean contact angles of the MgAlON-containing samples decreased slightly during the first 90 min of the holding period at 950 °C. This can be correlated with the settling of the metallic droplet, along with interfacial reaction processes. Considering that MgAlON synthesized at 1500 °C is reported to be a metastable phase at 950 °C, which is resistant to corrosion by metal melts, with interfacial reactions being kinetically hindered [10], it can be assumed that interface reactions can take place between the metal alloy and the amount of residual oxide phases in the ceramic substrate, such as MgO, $Al_2O_3$, AlN, and residual $MgAl_2O_4$ formed during reactive sintering.

Interfacial reactions, such as the reduction of residual $Al_2O_3$ or $MgAl_2O_4$ by the magnesium content of the alloy, were reported to reduce the contact angle through the induced structural and chemical changes at the interface [67].

During the remaining holding time, the respective contact angles stayed nearly constant, and none of the samples displayed contact angles lower than 144°, leading to the conclusion that no further interfacial reactions took place to significantly influence the contact angles and wetting behavior (cf. Figure 9).

One should note that the ceramic samples containing 76 wt% MgAlON (composition No. 4 I and II) displayed slightly higher contact angle values than the ones containing 99 wt% MgAlON (compositions No. 9 I and II, cf. Table 8 and Figure 9). Although the average difference between the values measured for these samples did not exceed 5°, a possible explanation can be seen in the conclusion drawn by Eustatopoulos et al., who stated that a lower amount of different phases at the interface lowers the surface energy and therefore results in an improved wettability, manifesting in lower contact angles [67].

Similar interfacial reactions were furthermore calculated and observed in previous wetting experiments by Schramm et al., using $Al_2O_3$, MgO, and MgAlON ceramic substrates and a capillary-purified AZ91 melt. MgAlON (99 wt%) was shown to be non-wettable, as well as non-reactive with the magnesium alloy melt at the common foundry temperature of 680 °C, with an average constant contact angle of 150° [76], leading to the conclusion that ceramic materials containing MgAlON can be regarded as promising materials for the filtration of light metals such as pure aluminum and magnesium and their alloys.

## 5. Conclusions

In this research work, technological, pressureless reaction sintering of MgAlON was conducted under flowing nitrogen at 1500 °C, using $Al_2O_3$, AlN, and MgO as starting materials. Based on previous research, compositions of 0.45–0.7 for $x_{Al_2O_3}$, 0.2–0.45 for $x_{MgO}$, and 0.1–0.35 for $x_{AlN}$ were selected, and thermodynamic calculations were performed. After sintering, the conversion degrees of the samples were evaluated through XRD analysis and the amount of nitrogen was measured for selected samples by means of carrier gas extraction.

Sessile drop tests with the aluminum alloy AlSi7Mg were conducted to evaluate the wettability and reactivity of selected sintered ceramic samples in order to evaluate the applicability of MgAlON-containing materials in light metal melt filtration.

The following conclusions can be drawn:

- Pressureless reactive sintering of $Al_2O_3$-AlN-MgO mixtures at a sintering temperature of 1500 °C with a duration of 3 h did not result in a near-equilibrium state and yields an average conversion degree to MgAlON of only 76 wt%.
- Reactive sintering of the same mixtures at 1500 °C for 6 h resulted in a near-equilibrium state, reaching an average conversion degree to MgAlON of 92 wt%.
- The highest conversion degrees of 99 wt% of MgAlON were reached in the MgO-rich corner of the quasi-ternary MgO-AlN-$Al_2O_3$ system with $Al_2O_3$:AlN ratios of around 20:6.
- At 1500 °C, MgAlON formed from $MgAl_2O_4$, which is a direct reaction product of MgO and $Al_2O_3$, and the subsequent, simultaneous dissolution of $Al_2O_3$ and AlN in the spinel structure. During this process, MgO was fully consumed after the first 3 h; the loss of magnesium due to evaporation or parasitic reactions was insignificant.
- The relative amount of nitrogen in the MgAlON phase of the samples with the lowest and highest conversion degree after 6 h of reactive sintering was measured to be 1.5 wt% for both samples.
- Contact angles measured between sintered MgAlON samples and AlSi7Mg via sessile drop tests at 950 °C were shown to be around 145–152°, indicating low wettability and non-reactivity, exceeding the ones measured for $MgAl_2O_4$, a material commonly used as a coating for metal melt filters. Therefore, MgAlON is regarded to be a promising ceramic material to be used in light metal melt filtration.

**Supplementary Materials:** The following supporting information can be downloaded at https://www.mdpi.com/article/10.3390/cryst12050654/s1. Table S1: Starting compositions of the powder mixtures and the weight fractions of individual phases in the samples sintered in nitrogen atmosphere for 3 and 6 h at 1500 °C. The compositions of the starting powders are depicted in Figure 5. The phase compositions of sintered samples were determined via XRD analysis (cf. Figure 6). Table S2: Lattice parameters of MgAlON formed in the powder mixtures No. 1-12 (cf. Figure 5 and Table 3) after 3 and 6 h of sintering at 1500 °C in nitrogen gas atmosphere that were determined using XRD, (cf. Figure 7). The measurement error for determined lattice parameters is approximately 0.0005 Å. Figure S1: Section of Rietveld refinements (CuK$\alpha$) of the powder mixture No. 12 ($x_{AlN} = 0.21$, $x_{MgO} = 0.22$, $x_{Al_2O_3} = 0.57$) before (as mixed) and after the sintering process for 3 h and 6 h at 1500 °C in reducing nitrogen atmosphere. Figure S2: XRD data of powder mixture No. 1 ($x_{AlN} = 0.35$, $x_{MgO} = 0.2$, $x_{Al_2O_3} = 0.45$) after the sintering process for 3 h (a) and 6 h (b) at 1500 °C in nitrogen atmosphere. Figure S3: XRD data of powder mixture No. 2 ($x_{AlN} = 0.27$, $x_{MgO} = 0.2$, $x_{Al_2O_3} = 0.53$) after the sintering process for 3 h (a) and 6 h (b) at 1500 °C in nitrogen atmosphere. Figure S4: XRD data of powder mixture No. 3 ($x_{AlN} = 0.18$, $x_{MgO} = 0.2$, $x_{Al_2O_3} = 0.62$) after the sintering process for 3 h (a) and 6 h (b) at 1500 °C in nitrogen atmosphere. Figure S5: XRD data of powder mixture No. 4 $x_{AlN} = 0.1$, $x_{MgO} = 0.2$, $x_{Al_2O_3} = 0.7$) after the sintering process for 3 h (a) and 6 h (b) at 1500 °C in nitrogen atmosphere. In addition, a comparison for the 6 h sample is given for the refinement of the spinel phase with and without the adjustment of cation vacancies (c). Figure S6: XRD data of powder mixture No. 5 ($x_{AlN} = 0.27$, $x_{MgO} = 0.28$, $x_{Al_2O_3} = 0.45$) after the sintering process for 3 h (a) and 6 h (b) at 1500 °C in nitrogen atmosphere. Figure S7: XRD data of powder mixture No. 6 ($x_{AlN} = 0.18$, $x_{MgO} = 0.28$, $x_{Al_2O_3} = 0.53$) after the sintering process for 3 h (a) and 6 h (b) at 1500 °C in nitrogen atmosphere. Figure S8: XRD data of powder mixture No. 7 ($x_{AlN} = 0.1$, $x_{MgO} = 0.28$, $x_{Al_2O_3} = 0.62$) after the sintering process for 3 h (a) and 6 h (b) at 1500 °C in nitrogen atmosphere. Figure S9: XRD data of powder mixture No. 8 ($x_{AlN} = 0.18$, $x_{MgO} = 0.37$, $x_{Al_2O_3} = 0.45$) after the sintering process for 3 h (a) and 6 h (b) at 1500 °C in nitrogen atmosphere. Figure S10: XRD data of powder mixture No. 9 ($x_{AlN} = 0.1$, $x_{MgO} = 0.37$, $x_{Al_2O_3} = 0.53$) after the sintering process for 3 h (a) and 6 h (b) at 1500 °C in nitrogen atmosphere. Figure S11: XRD data of powder mixture No. 10 ($x_{AlN} = 0.1$, $x_{MgO} = 0.45$, $x_{Al_2O_3} = 0.45$) after the sintering process for 3 h (a) and 6 h (b) at 1500 °C in nitrogen atmosphere. Figure S12: XRD data of powder mixture No. 11 ($x_{AlN} = 0.27$, $x_{MgO} = 0.22$, $x_{Al_2O_3} = 0.51$) after the sintering process for 3 h (a) and 6 h (b) at 1500 °C in nitrogen atmosphere. Figure S13: XRD data of powder mixture No. 12 ($x_{AlN} = 0.21$, $x_{MgO} = 0.22$, $x_{Al_2O_3} = 0.57$) after the sintering process for 3 h (a) and 6 h (b) at 1500 °C in nitrogen atmosphere.

**Author Contributions:** Conceptualization, C.G.A. and A.S.; methodology, M.T. and A.S.; software, M.T., O.F. and J.K. (Jakob Kraus); validation, M.T., O.F., J.K. (Jakob Kraus) and S.B.; formal analysis, investigation M.T., A.S. and S.B.; resources A.S.; writing—original draft preparation, A.S.; writing—review and editing, M.T., C.S., D.R., J.K. (Jens Kortus), J.K. (Jakob Kraus) and S.B.; project administration, C.G.A. All authors have read and agreed to the published version of the manuscript.

**Funding:** This research was funded by the German Research Foundation (DFG), Collaborative Research Center 920 "Multifunctional Filters for Metal Melt Filtration—A contribution towards Zero Defect Materials", subprojects A03, A04, A05, and C06 (project ID 169148856). Furthermore, the authors thank the ZIH in Dresden and the URZ in Freiberg for the computational time and support. The computations at the URZ were performed on the compute cluster of the Faculty of Mathematics and Computer Science of the TU Bergakademie Freiberg, funded by the DFG—project ID 397252409.

**Acknowledgments:** The authors would like to thank Beate Kutzner for the carrier gas extraction measurements.

**Conflicts of Interest:** The authors declare no conflict of interest. The funders had no role in the design of the study; in the collection, analyses, or interpretation of data; in the writing of the manuscript, or in the decision to publish the results.

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
