# Peer review of "Reaction Sintering of MgAlON at 1500 °C from Al2O3, MgO and AlN and Its Wettability by AlSi7Mg"

_crystals, doi:10.3390/cryst12050654_

Round 1

Reviewer 1 Report

The authors reported detailedly the synthesis of MgAlON by solid-state reaction at 1500℃, thermodynamic calculation of the system MgO-Al2O3-AlN and evaluation of wetting behaviors of MgAlON ceramics by liquid AlSi7Mg alloy. In a whole, the work is excellent. Therefore, I recommend for publication in Crystal after minor revision.

1) On Line 120-121, heterovalent substitution causes a formation of vacancies in cationic sites?  It appears that vacancies should decrease when N3- substitute O2- in gamma-AlON.

2) In Fig. 2b and 3b, MgO would possibly destabilize when the partial pressure of O2 were reduced to extremely low concentration. Therefore, relevant explainations for this were necessary if MgO+AlN regions were still located on the bottom right side in Fig. 2b and 3b. Alternatively, plot an extra equilibrium line.

3) Unit of gas pressure is Bar or Pa in this work? Please check.

Author Response

Dear Reviewer 1,

Thank you very much for your time and effort. We tried to make the draft better by taking your comments and advice into account. Please, see the attachment. Excuse me, that in this document all responses to the reviewers are included.

Kind regards

Christiane Scharf 

Reviewer 2 Report

The manuscript dealt with the preparation and mechanism of MgAlON by reaction sintering at 1500 °C from Al2O3, MgO and AlN. The wettability of MgAlON by AlSi7Mg at 950°C was evaluated and compared with that of MgAl2O4 and Al2O3. Overall the manuscript was well organized, but it can also be improved.

1. In Page 1 Line 39-40, “Bending strengths up to 25 MPa at 1000 °C were reached in MgAlON spinel composites [11].” This is not comprehensive, since the flexural strength of Mg0.27Al2.58O3.73N0.27 transparent ceramic at 1000 °C and 1200 °C has already been determined to be 200 MPa and 125 MPa, respectively. (c. f. ZHANG Zhou, WANG Hao, TU Bing-Tian, XU Peng-Yu, WANG Wei-Min, FU Zheng-Yi. Characterization and Evaluation on Mechanical Property of Mg0.27Al2.58O3.73N0.27 Transparent Ceramic[J]. Journal of Inorganic Materials, 2018, 33(9): 1006-101.) 

2. In Page 10 Line 333-334, “For the Rietveld refinement of the oxynitride spinel phase (MgAlON), the structure of MgAl2O4 was employed (see Table 4).” Actually, the research for determining the crystal structure of some MgAlON has been conducted in detailed. (c. f. Xiao Zong, Lu Ren, Hao Wang, Bingtian Tu, Weimin Wang, and Zhengyi Fu. Structural Study of MgyAl(8+x–2y)/3O4–xNx (0 < x < 0.5, 0 < y < 1) Spinel Probed by X-ray Diffraction, 27Al MAS NMR, and First-Principles Calculations. Inorganic Chemistry 2020 59 (23), 17009-17017.) I do think these determined crystal structures of complex MgAlON spinel solid solution can help the authors to improve their Rietveld refinement.

Author Response

Dear Reviewer 2,

Thank you very much for your time and effort. We tried to make the draft better by taking your comments and advice into account. Please, see the attachment. Excuse me, that in this document all responses to reviewers are included.

Kind regards

Christiane Scharf

Reviewer 3 Report

The manuscript “Reaction Sintering of MgAlON at 1500 °C from Al2O3, MgO 2 and AlN and its wettability by AlSi7Mg” by Alina Schramm et al. is devoted to systematical investigation of a technological synthesis of MgAlON. The manuscript is supported by number of diagrams of ternary systems. Also, a lot of data is based on the XRD analysis. However, there are no XRD patterns were presented in the text or in the Supporting Information. It must be shown.

Also, there are several phrases:

Page 10 line 33: For the Rietveld refinement of the oxynitride spinel phase (MgAlON) – where is the the data for the Rietveld refinement?

Page 12 line 385: XRD analysis showed… - no data were provided.

So, these data is nessesury.

There are a few small comments:

Page 3 line 108: Wyckoff sites – the letter near the number must be in italic font

Fig. 9 must be improved

Table S2: the Lattice parameters must be given along with the errors of the determination.

Author Response

Dear Reviewer 3,

Thank you very much for your time and effort. We tried to make the draft better by taking your comments and advice into account. Please, see the attachment. Excuse me, that in this document all responses to reviewers are included.

Kind regards

Christiane Scharf 

Round 2

Reviewer 3 Report

The authors have provided all the required data. The manuscript can be accepted for publication in the present form.